# Modulation of Organogenesis and Somatic Embryogenesis by Ethylene: An Overview

**DOI:** 10.3390/plants10061208

**Published:** 2021-06-14

**Authors:** Mariana Neves, Sandra Correia, Carlos Cavaleiro, Jorge Canhoto

**Affiliations:** 1Center for Functional Ecology, Department of Life Sciences, University of Coimbra, 3000-456 Coimbra, Portugal; mariananevespt@gmail.com (M.N.); sandraimc@ci.uc.pt (S.C.); 2CIEPQPF, Faculty of Pharmacy, University of Coimbra, Pólo das Ciências da Saúde, Azinhaga de Santa Comba, 3000-548 Coimbra, Portugal; cavaleir@ff.uc.pt

**Keywords:** ethylene biosynthesis, ethylene inhibitors, in vitro culture, plant hormones, S-adenosylmethionine, stress responses

## Abstract

Ethylene is a plant hormone controlling physiological and developmental processes such as fruit maturation, hairy root formation, and leaf abscission. Its effect on regeneration systems, such as organogenesis and somatic embryogenesis (SE), has been studied, and progress in molecular biology techniques have contributed to unveiling the mechanisms behind its effects. The influence of ethylene on regeneration should not be overlooked. This compound affects regeneration differently, depending on the species, genotype, and explant. In some species, ethylene seems to revert recalcitrance in genotypes with low regeneration capacity. However, its effect is not additive, since in genotypes with high regeneration capacity this ability decreases in the presence of ethylene precursors, suggesting that regeneration is modulated by ethylene. Several lines of evidence have shown that the role of ethylene in regeneration is markedly connected to biotic and abiotic stresses as well as to hormonal-crosstalk, in particular with key regeneration hormones and growth regulators of the auxin and cytokinin families. Transcriptional factors of the ethylene response factor (ERF) family are regulated by ethylene and strongly connected to SE induction. Thus, an evident connection between ethylene, stress responses, and regeneration capacity is markedly established. In this review the effect of ethylene and the way it interacts with other players during organogenesis and somatic embryogenesis is discussed. Further studies on the regulation of ERF gene expression induced by ethylene during regeneration can contribute to new insights on the exact role of ethylene in these processes. A possible role in epigenetic modifications should be considered, since some ethylene signaling components are directly related to histone acetylation.

## 1. Introduction

Micropropagation has been studied and developed through the years, and is currently one of the best alternatives for large-scale propagation and conservation of a wide range of plant species [1]. In practical terms, micropropagation is the culture of plant cells, tissues, or organs in a well-defined culture medium, under controlled aseptic conditions [2]. Several in vitro culture methods have been developed and, in combination with molecular techniques, have allowed for the multiplication of selected genotypes with better resistance to diseases and stress tolerance capacity [1]. Advances in plant tissue culture have also provided a deep analysis of several developmental plant processes and have helped to understand some of their molecular mechanisms [3]. 

Apart from the genotype and the type of explant, nutritional and environmental conditions are critical factors to achieve an optimal response from plant tissues. Mineral composition, carbon source, plant growth regulators (PGRs), gelling agents, pH, light, temperature, and also the gaseous environment are the factors to take into consideration for the success of the different in vitro plant propagation and regeneration processes [1].

Plant regeneration systems, such as organogenesis and somatic embryogenesis (SE) are well-known micropropagation processes based on plant cells pluripotency/totipotency. In organogenesis, plant organs, such as shoots, roots, and even flowers, can be formed from cultured explants. However, for micropropagation purposes, the most interesting is de novo shoot meristem formation followed by shoot growth and rooting. SE is a more complex developmental pathway by which bipolar structures identical to zygotic embryos are developed from somatic cells through a complex dedifferentiation process, followed by totipotency acquisition and formation of somatic embryos [3,4]. The presence of PGRs in the culture media—in particular, auxins and cytokinins—is required for the success of plant regeneration, since a balance between both hormones contributes to enhance the in vitro regenerative capacity of a myriad of plant species [3]. While these PGRs are deliberately placed in the culture medium in a well-defined added concentration, and a dose–response assay can be easily evaluated, the role of gaseous compounds, such as the hormone ethylene, is more difficult to analyze. This gas is usually liberated during the in vitro culture of plant tissues accumulating in the surrounding atmosphere when closed vessels are used. Therefore, its presence and consequent accumulation in culture vessels is a consequence of the method and not of a deliberate addition [5]. Thus, understanding whether ethylene influences in vitro culture is relevant for the improvement of in vitro regeneration processes and somehow may explain why some species or tissues seem to be more recalcitrant than others. 

Taking this into consideration, this review presents a general overview of the ethylene effect on the main in vitro regeneration processes, revising and discussing the studies that have been published over the years. A brief overview of ethylene’s biosynthesis and signaling pathways and its different modulations are also presented herein. 

## 2. Ethylene: Biosynthesis and Signaling Pathway 

The plant hormone ethylene, C_2_H_4_, is a simple, small, and gaseous molecule synthetized and released by plant tissues. Despite its simplicity, this hormone has a crucial role in controlling several physiological and developmental processes, ranging from plant growth to fruit ripening [6]. Its biosynthesis was intensively studied in the second half of the 20th century, the period in which methionine was found as an ethylene precursor by Lieberman et al. [7]. Later on, Adams and Yang [8,9] identified and demonstrated that s-adenosylmethionine (SAM) and 1-aminocyclopropane-1-carboxylic acid (ACC) were the intermediates in the conversion of methionine to ethylene. The same authors also proposed that the methionine involved in ethylene production was recycled by a set of reactions termed the methionine cycle, nowadays known as the Yang cycle. The identification of the ethylene precursors was the major breakthrough for the establishment of the ethylene biosynthetic pathway, as it is known today. In a general overview, ethylene biosynthesis begins with methionine, which is converted to ethylene through three successive enzymatic reactions (Figure 1). In the first reaction, methionine is converted to SAM by SAM synthetase (SAMS), then SAM is converted to ACC by ACC synthase (ACS), and finally ACC oxidase (ACO) degrades ACC, releasing ethylene [10].

The effect of this hormone on plant tissues is the result of a complex subsequent signaling pathway in which some molecular mechanisms are not yet fully understood. Diverse studies with *Arabidopsis thaliana* allowed the identification and characterization of the signaling ethylene components, such as the ethylene receptors. Five ethylene receptor isoforms have been identified, denominated ETHYLENE RESISTANT1 (ETR1), ETR2, ETHYLENE INSENSITIVE4 (EIN4), ETHYLENE RESPONSE SENSOR1 (ERS1), and ERS2. Despite some structural differences, all the isoforms are homodimers, constituted by two monomers, stabilized by two disulfide bonds. The ethylene-binding domain is highly conserved at the N-terminus. This ethylene-binding site is formed by three transmembrane α-helices of each monomer, that together coordinate at least a copper ion, Cu(I) (see [11] for a detailed characterization). ETR1 in *Arabidopsis* was not only the first ethylene receptor identified but also the first plant hormone receptor identified and cloned [12,13]. Further studies with ETR1 have allowed the identification of key components responsible for the high affinity of this hormone to its receptors. Therefore, it is known that Cu(I) has a fundamental role since it acts as a cofactor, mediating the binding of ethylene to its receptors [14]. All the receptor isoforms coordinate Cu(I) in their transmembrane portion (ethylene-binding site). In the absence of this cofactor, ethylene binding is compromised and, consequently, this hormone is not perceived by plant tissues [11].

Recently, Binder [15] reviewed the canonical ethylene signaling pathway in *Arabidopsis* and suggested that ethylene signal transduction may be even more complex. To summarize—canonically, ethylene, once biosynthesized, acts as an inverse agonist, binding and inhibiting its receptors in the endoplasmic reticulum (ER) membrane. Downstream, ethylene receptors interact with the serine/threonine-protein kinase CONSTITUTIVE TRIPLE RESPONSE1 (CTR1), inhibiting its activity. Therefore, CTR1 can no longer phosphorylate EIN2, leading to a reduction in EIN2 ubiquitination and consequently resulting in an increase of EIN2 levels. The lack of phosphorylation also promotes the cleavage of the EIN2 C-terminus (EIN2-C) from the N-terminus by an unknown protease. EIN2-C functions as a positive regulator, moves from the ER to the nucleus, increasing the levels and the activity of the transcription factors EIN3, ETHYLENE INSENSITIVE3-LIKE1 (EIL1) and EIL2, which lead to transcriptional changes and consequently promote ethylene responses [15,16]. As the ethylene signaling pathway is not the main focus of this review it will not be discussed in detail, but for readers who desire detailed information, recent reviews by Ju and Chang [16] and Binder [15] are recommended. A brief outline on ethylene biosynthesis and consequent signaling is present herein (Figure 1).

## 3. Culture Vessels and Ethylene Accumulation 

The use of closed culture vessels to prevent contaminations is one of the requirements of in vitro culture. Test tubes, glass vials, Petri dishes, and boxes made of polycarbonate or polypropylene are some examples of vessels used in micropropagation. These vessels can then be closed with different types of closures, such as screw lids, polycarbonate or polypropylene lids, plastic film, Parafilm, and cotton plugs, among others. The different types of vessels and, in particular, the different possible closures allow a greater or lesser gas exchange rate with the external environment [17,18] without compromising aseptic conditions. 

To detect and measure ethylene, several methods can be used, such as gas-chromatography (GC), electrochemical sensing, and optical detection. All of these detection methods have their advantages and limitations. Therefore, the selection of the most suitable method will depend on the purpose of each study. For instance, when sensitivity and real-time detection is needed, optical detection is recommended. Whether, in addition to ethylene, other gases need to be quantified, GC will be the most suitable method [19]. GC is the oldest and the most commonly used method of ethylene detection according to the literature, with its first application reported by Huelin and Kennett [20]. In in vitro studies using GC to measure ethylene (e.g., ethylene accumulation in culture vessels or the rate of ethylene production by plant tissues), sealed vessels with a rubber septum are required. This type of closure allows the withdrawal of gas samples from the headspace, which can then be detected and analyzed [5]. 

Most vessels currently used in micropropagation seem to allow gas exchanges. This aspect prevents the excessive accumulation of ethylene and other gases in the culture atmosphere. However, when exogenous treatments of ethylene are performed, its loss should be considered, especially in studies entailing longer culture periods. Jackson et al. [21] studied the ventilation capacity of different vessels and sealing methods, monitoring the rate loss of ethylene previously injected. For instance, in the same type of vessel (polycarbonate box with a polypropylene lid), vessels that were loosely closed (i.e., using only the respective lid of the vessel) lost half of the injected ethylene in 1–2 h, while tightly closed vessels (i.e, using two layers of Nescofilm between the join of the vessel and the lid) increased the half-time of ethylene depletion to 16 h. Glass vessels with glass lids were also tested. Whereas the loose sealing presented the same results as the other vessels (approximately 2 h), the tight sealing increased the half-time of the ethylene depletion to 100 h. The authors also monitored the accumulation of ethylene produced from plant tissues during 28 days. Ethylene accumulated in all vessels, with a greater accumulation in tightly closed vessels, as expected, and seemed to decrease during the last days of the culture. 

The amount of ethylene accumulated in the atmosphere of culture will depend on both the gas exchange rate between the vessel and the exterior and the rate of ethylene production by plant tissues themselves [5]. Note that even though these gas exchanges occur and thus some ethylene is lost, its accumulation in culture vessels persists, to a greater or lesser extent, and may be a limiting factor for the success of in vitro culture; in particular, for the most ethylene-sensitive species. In this regard, understanding the effect of this hormone in plant tissues can be assessed by a chemical or a genetic approach, a subject that will be reviewed in the next two sections. 

## 4. Chemical Modulation of Ethylene Responses 

All the knowledge acquired regarding ethylene biosynthetic and signaling pathways have allowed both modulation of its biosynthesis and perception in plant tissues, therefore enhancing, decreasing, or inhibiting ethylene responses. Manipulating these responses is a crucial step to understanding the influence of this hormone in a myriad of physiological processes. In this context, several chemical inhibitors and enhancers of ethylene action have been studied and widely used. 

Regarding chemical inhibitors, they can be divided into two main categories, depending on whether they inhibit its biosynthetic pathway or its signaling pathway [22]. 

ACS and ACO enzymes are the key targets in the inhibition of ethylene biosynthesis. Compounds such as aminoethoxyvinylglycine (AVG) and aminooxyacetic acid (AOA) are the main ACS inhibitors used through the diverse ethylene studies. These two compounds revealed an effective inhibition of ACS activity with significant reductions in the production of ACC and, consequently, ethylene [23,24,25]. ACS is a pyridoxal phosphate-dependent enzyme for the conversion of SAM to ACC [24] and both AVG and AOA are inhibitors of pyridoxal phosphate-mediated reactions [24,26]—the reason for the great effectiveness of this inhibition. However, Schaller and Binder [22] stressed the non-specificity of both compounds and their undesirable off-target effects. AVG and AOA inhibit other pyridoxal phosphate-dependent enzymes, such as tryptophan aminotransferase, involved in indole-3-acetic acid (IAA, endogenous auxin) biosynthesis [27]. 

Regarding the ACO enzyme, cobalt ions, Co(II), and aminoisobutyric acid (AIB) are known for inhibiting its activity [25,28,29]. ACO is a member of the 2-oxoglutarate-depedent dioxygenase superfamily, which depends on Fe(II) as a cofactor to convert ACC into ethylene [30]. Co(II) is reported as an effective inhibitor of ACO activity due to its competitive inhibition with Fe(II) for the active site [31]. AIB, on the other hand, acts as an analogue of ACC, competitively inhibiting the conversion of ACC into ethylene [28]. Salicylic acid also inhibits ACO activity [32] and for this reason has been also used as an ethylene modulator, albeit to a lesser extent.

Due to the non-specific effect of some compounds, the demand for specific ethylene biosynthesis inhibitors has increased. Therefore, novel inhibitors of ACS and ACO are being studied using a phenotype-based screening approach in *Arabidopsis* [33,34], identifying small chemical compounds that specifically inhibit these enzymes.

The main target of ethylene signaling inhibitors are ethylene receptors [22]. One of the strategies to inhibit ethylene action in plant tissues is the use of silver ions, Ag(I), usually applied as silver nitrate (AgNO_3_) or silver thiosulfate (STS). Ag(I) is a widely used potent inhibitor of ethylene perception [35], due to the replacement of the Cu(I) cofactor required for ethylene binding, thus blocking ethylene signaling [36]. Despite the highly conserved ethylene-binding domain among all ethylene receptors isoforms [11], Ag(I) seems to replace Cu(I) mainly at ETR1, however, this inhibition is sufficient to inhibit ethylene responses [37]. Despite being one of the most common inhibitors used, silver has some phytotoxicity already recognized. Silver ions, applied as AgNO_3_, induced oxidative stress in seedlings of mustard [38] and in leaves and roots of tobacco plants [39]. Furthermore, treatments with AgNO_3_ reduced some physiological parameters, such as chlorophyll a content and plant biomass in wheat [40] and *Spirodela polyrhiza* plants [41]. In a similar way to AVG and AOA, silver also has off-target effects [22], increasing auxin efflux [42]. 

Gaseous compounds such as 1-methylcyclopropene (1-MCP), trans-cyclooctene (TCO) and 2,5-norbornadiene (NBD) also can be used as inhibitors of ethylene receptors [22]. These olefin compounds are competitive inhibitors, competing with ethylene for its receptors. From the three compounds, 1-MCP is the best alternative to control ethylene responses due to its effective inhibition at extremely low concentrations, prolonged effect and non-toxicity, triggering lower possible side-effects [43]. These characteristics make 1-MCP one of the best compounds to apply in ethylene studies due to its lower rate of off-target effects [22]. Due to the gaseous nature of these receptor inhibitors, the use of gas-tight chambers is required for their maintenance in the culture atmosphere (see [22] for chambers examples). 

It should be noted that the use of inhibitors of ethylene biosynthesis compared with inhibitors of ethylene signaling do not protect against exogenous sources of ethylene. This fact can offer an advantage by allowing the reversion of their effect in plant tissues. For instance, the application of exogenous ethylene to the culture atmosphere, and if only ACS was inhibited, exogenous treatments with ethylene’s intermediate ACC can also be used [22].

Apart from the use of both classes of inhibitors mentioned above, changing the ethylene atmospheric concentration is another relevant strategy, particularly in in vitro tissue cultures due to the use of closed vessels. In this regard, pure ethylene or its precursors can be applied in order to enhance ethylene action in plant tissues [5]. The ethylene precursor ACC has been widely used in vitro, since it promotes an increase of ethylene biosynthesis by ACO, enhancing ethylene production [44]. Another ethylene precursor, 2-chloroethylphosphonic acid, commonly known as ethephon (ETH), has been proven to cause responses similar to those of exogenous ethylene treatments effectively [45] and, for this reason, has also been widely used throughout several studies. Methylglyoxal bis(guanylhydrazone) (MGBG) is also used to promote ethylene biosynthesis, albeit to a lesser extent, due to inhibition of polyamine biosynthesis. Polyamines synthesis use SAM as precursor and its inhibition stimulates ethylene production [46]. On the other hand, it is also possible to remove ethylene from culture atmosphere using compounds such as mercuric perchlorate (HgClO_4_) and potassium permanganate (KMnO_4_, [5]). Mercuric perchlorate forms complexes with ethylene [47], whereas potassium permanganate oxidizes ethylene [48]. Both act as ethylene absorbents, therefore decreasing the ethylene concentration in the culture atmosphere and, consequently, the action of ethylene on plant tissues. A schematic diagram of some of the principal ethylene modulators and their points of action is presented herein (Figure 2).

## 5. Genetic and Epigenetic Modulations of Ethylene Responses 

Aside from the chemical modulation previously reviewed, the development of molecular techniques has helped us to understand and modulate the effect of this hormone on plant tissues while contributing to detailed insights on the role of each component of the ethylene biosynthesis and signaling pathways. With genetic engineering it has been possible to alter ethylene levels in plant tissues via either or both the ACS and ACO enzymes, or its perception through the components of its signaling [49]. Studies using plant mutants with altered ethylene sensitivity or production [50,51] as well as transgenic plants with altered ethylene biosynthesis or perception [49] have been important contributions for the characterization of ethylene signaling and its responses as they are currently known. 

Different mutants in *Arabidopsis*, with different mutations regarding the components of ethylene biosynthesis or signaling pathways, are well characterized. Depending on the type of mutation, the plant mutant can exhibit distinct phenotypes, such as ethylene insensitivity [13], ethylene constitutive responses (i.e., exhibiting the phenotype observed in plants treated with ethylene [52]), and ethylene overproduction [53]. For instance, the *etr1-1* mutant, characterized by a missense mutation at the *ETR1* gene, exhibits insensitivity to ethylene. This mutation alters a single amino acid at the N-terminus of the ETR1 receptor in the ethylene-binding domain [12,13] and for this reason eliminates the capability to bind ethylene [54], comparable to what happens in Ag(I) treatments. Similar phenotypes are exhibited in both *etr2-1* and *ein4-1* mutants, with similar mutations to that of *etr1-1* in the *ETR2* and *EIN4* genes, respectively [55,56]. These similarities seem to be in agreement with the conservation of the ethylene-binding domain among the receptor’s isoforms. On the other hand, the *ctr1-1* mutant exhibits an ethylene constitutive response. This mutant is characterized by a single amino acid substitution in the serine/threonine-protein kinase CTR1 at the highly conserved kinase residues, resulting in a disruption of its catalytic domain [52] and consequently inactivating its function. Regarding the ethylene biosynthesis, the recessive mutant *eto1* and the dominant mutants *eto2* and *eto3* are characterized by increasing the stability of ACS, exhibiting an ethylene overproduction phenotype [50,53,57]. These ethylene-overproducing mutants are characterized by mutations affecting the posttranscriptional regulation of ACS5 and ACS9 isoforms, resulting in an increase of ethylene biosynthesis [53]. 

Beyond the mutants described, modulating the amount of ethylene produced or its perception, through the creation of transgenic lines in several species, has been a goal of genetic engineering, particularly for commercial purposes [49] due to the effect of excessive amounts of ethylene in fruit ripening and senescence. Therefore, transgenic approaches have been applied using diverse molecular techniques, which have allowed the silencing or the overproducing of the enzymes involved in ethylene biosynthesis, leading to different accumulations of this hormone in plant tissues. Most of these strategies entail the downregulation of *ACS* or *ACO* expression, leading to a reduction of its enzymatic activity and consequent reduction of ethylene production [30]. This modulation can be achieved using sense- or antisense-mediated gene silencing, co-suppression, or by using an RNA interference (RNAi) approach, with several studies carried out in several plant species [30,49,58,59]. Regarding an ethylene overproduction strategy, the overexpression of the *ACO* gene was achieved in poplar [60] and in safflower [61], by transforming these plants with an overexpression vector containing the open reading frame of the specific ACO. Note that the ACO enzyme is the most studied and suitable target regarding genetic alterations of ethylene biosynthesis due to its reduced risk of interfering with other metabolic pathways, such as the methionine cycle and ACC metabolism [30]. 

A promising strategy to alter ethylene responses is the use of the genome editing system CRISPR/Cas9, a potent system capable of generating efficiently mutations in specific targeted genes in plants [62]. In fact, very recently, Xu et al. [63] reported successful generation of *ACO*-edited mutants in petunia, which exhibited lower ethylene production through the editing of the specific *ACO* gene using the CRISPR/Cas9 system. 

Currently, using plants with these genetic alterations can give us a more specific approach to the effects of this hormone regarding both in vivo and in vitro processes, with a decrease of possible off-targets. Therefore, these plants, especially the *Arabidopsis* mutants, are currently important models in the study of the impact of ethylene in the success of several in vitro processes. However, the use of transgenic or mutant plants involves more extensive and expensive protocols, compared to the use of chemical inhibitors. A chemical approach is easier to manipulate and its effect can be also easily reverted (e.g., changing plant tissues to a non-treatment medium). Furthermore, these chemicals can be easily applied only at specific points of each in vitro process. 

Epigenetic regulation in plants is associated with plant plasticity and the plant’s ability to survive in response to stress. Variations in gene expression by DNA methylation, histone modification, and noncoding RNAs are directly associated with epigenetic modifications [64]. Studies related to the involvement of ethylene on epigenetic processes in plants are scarce; however, this line of research needs to be exploited. The precursor of ethylene biosynthesis, SAM, is the donor of methyl groups for DNA methylation [65,66], and ethylene signaling seems to be linked to chromatin regulation by histone modifications directly mediated via EIN2 [67]. In animal cells [68], the levels of metabolic substrates such as SAM and S-adenosylhomocysteine (SAH), the derived compound resulting from the transfer of the methyl group of SAM to a donor, change the expression of genes related to the growth and health. In plants [69], ethylene can be an extra player in this balance since the pool of SAM controls ethylene production. For instance, the knockdown of *SAMS* genes in rice, characterized by a decrease of SAM synthesis and consequently ethylene production, showed a great reduction in DNA and histone methylation, leading to a delayed germination, reduced fertility, and late flowering [70]. The role of epigenetic modifications on plant regeneration efficiency, with dynamic changes in chromatin structure leading to callus formation has been recognized [71]. Recent studies reported a direct role of the positive regulator of ethylene signaling, EIN2-C, in histone acetylation [72,73,74,75]. EIN2-C seems to be involved in the recruitment of an unknown histone acetyltransferase (HAT) and simultaneously in the inhibition of histone deacetylases (HDAs) activity, leading to acetylation of specific histone residues and subsequent EIN3-dependent transcriptional activation [67]. Overall, ethylene and/or its modulation may affect the activity of important enzymes linked to epigenetic modifications, such as methyltransferases, which catalyze methylation of a large spectra of substrates such as DNA and RNA, and HATs and HDAs, linked to histone modifications. Thus, modifying transcription and translation processes and the subsequent control of mechanisms such as plant growth and responses to biotic and abiotic stresses [76].

## 6. Ethylene Integration in Hormone and Stress-Induced In Vitro Plant Regeneration 

Plant regeneration depends on cellular plasticity, i.e., the ability of cells to assume different differentiation pathways, ending in different cellular fates [3]. The regeneration capacity differs from species to species and, within the same species, different explants can show different response to regeneration [77]. Juvenile explants, such as young leaves, or immature zygotic embryos, generally have high generation capacity [3]. In vitro plant regeneration can occur directly from explant or indirectly through a preinduced pluripotent mass of cells, denominated callus [4]. A balance between auxins and cytokinins has a fundamental role in promoting organogenesis, inducing callus formation and subsequent shoot development [78,79,80]. SE is mainly triggered by auxins and by stress factors, such as osmotic and/or oxidative stress, leading to acquisition of embryogenic competence [81,82]. The crosstalk between ethylene with auxins and cytokinins has been progressively unveiled. In a brief overview, Swarup et al. [83] have reported an increase in the rate of the endogenous auxin synthesis, IAA, in *Arabidopsis* seedlings in response to 10 and 100 µM ACC. A decrease in IAA content in the presence of 10 µM AVG was also observed, suggesting that ethylene upregulates auxin biosynthesis. However, note that AVG itself can also reduce IAA biosynthesis by inhibition of tryptophan aminotransferase as already mentioned [27]. In *Arabidopsis* roots, an increase of IAA concentration in the presence of ACC (100 µM) and a decrease of IAA synthesis rate at 10 µM AVG have also been reported [84]. A reduction in free IAA content in the roots of the ethylene-overproducer mutant *eto1-1* was reported, which was also found, to lesser extent, when wild-type *Arabidopsis* was submitted to 1 μM ACC [85]. Furthermore, a recent work [86] showed also lower levels of free IAA in the same mutant, due to a downregulation of key auxin biosynthetic genes. Ethylene seems also to regulate auxin transport. ACC (1 µM) treatments increased the expression of PIN-FORMED3 (PIN3) and PIN7 auxin efflux carriers in *Arabidopsis* roots while AVG (1 µM) reduced this expression, leading to an inhibition of lateral root formation/growth since it prevents localized auxin accumulation in the lateral root-forming zone [85]. Moreover, in *Arabidopsis* seedlings, ethylene treatments increased AUX1 influx carrier expression with auxin accumulation at the apical hook. In addition, an increase in auxin biosynthesis at the inner side of the hook was reported, leading to an enhanced curvature [87]. Overall, ethylene seems to modulate auxin transport/distribution through PINs and AUX1 transporters (see [88] for a detailed crosstalk). Regarding auxin’s effect in ethylene biosynthesis, exogenous IAA treatment (20 µM) applied to *Arabidopsis* seedlings enhanced the constitutive expression of diverse *ACS* gene family members in roots [89]. *ACS1* and *ACS2* genes were also upregulated in the presence of 50 and 100 µM IAA in watermelon leaves [90]. Concerning cytokinins, *Arabidopsis* seedlings treated with 5 µM of the synthetic cytokinin, benzyladenine (BA), showed a higher ethylene production due to the stabilization of ACS5 isoform [53]. Similarly, it was found that BA treatments increase the stability of ACS5 and ACS9 isoforms in *Arabidopsis* [91], suggesting that cytokinin acts post-transcriptionally, leading to ACS stabilization and a subsequent increase in ethylene production. Ethylene seems also to increase the levels of cytokinins modulated by an upregulation of key cytokinin biosynthetic genes, as reported in the *Arabidopsis* mutant *eto1-1* [86].

Apart from the role of plant hormones in regeneration, wounding—initially applied to explants—seems to be essential to initiate both regeneration processes [3]. Wounding triggers stress responses, inducing transcriptional changes with marked changes in metabolism and protein synthesis. In turn, inherent regulators of the cell cycle are activated, leading to cell proliferation and consequent callus formation [92]. Wounding is also crucial to increase the levels of SE responsiveness in the leaves of *Solanum betaceum* [93] and *Arbutus unedo* [94]. Interestingly, ethylene biosynthesis seems to be stimulated by wounding in several plant tissues [95,96,97,98]. Wounding seems to enhance ACS activity over time after cutting, leading to a greater ACC synthesis and consequently increases ethylene production [95]. This increase seems to be in agreement with the role of ethylene in several stress responses, since its role in both biotic [99,100,101] and abiotic [102,103] stresses has been recognized. Furthermore, wounding also promotes an accumulation of cytokinins at cutting sites leading to callus formation [92], which can also be related to the increase in ethylene biosynthesis [53,91]. 

As the success of regeneration processes highly depends on hormonal and stress induction, the influence of ethylene in key regeneration hormones, such as auxins and cytokinins, and in stress responses, should be markedly considered.

## 7. Influence of Ethylene Modulation in Regeneration Processes 

The influence of ethylene in in vitro processes started to be studied more deeply in the last years of the 20th century, with relevant reviews published by Biddington [5] and Kumar et al. [104]. More recently, numerous studies have been carried out regarding regenerating processes, such as axillary meristem culture, organogenesis, and SE. In the next sections an overview of those studies will be presented and discussed in order to elucidate the effect of this hormone in in vitro regeneration. Detailed summaries of several studies using chemical modulators or using mutants and transgenic lines are presented in Table 1 and Table 2, respectively. 

### 7.1. Organogenesis

Ethylene modulation seems to have different impacts in the organogenesis process, depending on the species. In gloxinia, inhibiting ethylene perception using silver thiosulfate, increased the number of shoots per leaf explants [105]. Furthermore, inhibition of ethylene biosynthetic enzymes, with AVG (6.24 µM) and cobalt chloride (7.7 µM), also increased the number of shoots per explant, albeit to a lesser degree when compared to silver thiosulfate treatments. However, at 62.43 and 124.87 µM AVG or 77 and 154 µM CoCl_2_, both chemicals showed a negative impact on regeneration. Inhibition of ethylene perception seems also to positively affect shoot regeneration from cotyledons in melon [106]. Regeneration capacity has increased by about twofold using 60 or 120 µM of silver nitrate. In turn, shoot regeneration capacity decreased in the presence of 69.2 and 138.4 µM ETH treatments, where ethylene production was significantly higher compared to both control and silver nitrate treatments. In agreement, using the leaves and cotyledons of a melon transgenic line (Table 2), expressing antisense *ACO*, greatly increased shoot regeneration [107]. Leaves and cotyledons show a 3.5- and 2.8-fold increase in regeneration capacity when compared to wild-type, respectively. Despite a great decrease of ACO activity in both transgenic explants, cotyledons showed a higher ethylene production compared with leaves, which can justify the differences observed in the regeneration response of both transgenic explants [107]. Explants of wild-type or transgenic lines treated with ETH showed a highly decrease in regeneration capacity and, at the higher ETH concentration (100 µM) a complete inhibition of regeneration was reported. A considerable increase of shoot regeneration capacity from hypocotyls of mustard in the presence of 17.66 µM silver nitrate was also detected [108]. Also in mustard, both ethylene inhibitors AVG and silver nitrate enhanced the shoot regeneration in leaf discs and petioles [109], whereas combined treatments with ethylene inhibitors and ETH decreased the regeneration capacity in leaf explants. Pretreated leaf explants with silver nitrate also showed a decrease in shoot regeneration capacity in further organogenesis induction with silver nitrate. However, shoot regeneration capacity of the same explants was not affected in control or in AVG treatments [109].

Furthermore, pretreated plants with AVG maintained the same regeneration capacity in posterior induction with both inhibitors. Authors suggest that mustard plants can be more sensible to Ag (I) and to its accumulation than to AVG. Actually, in the same study, further plant growth parameters were negatively affected by silver nitrate, but not by AVG treatment. Moreover, as mentioned in Section 4, mustard seedlings exposed to AgNO_3_ (1 and 3 mM) exhibited higher levels of oxidative stress along with a reduction in photosynthesis, due to silver nitrate accumulation, also resulting in a decline of plant growth parameters [38]. Shoot regeneration from leaf discs or hypocotyl segments of transgenic lines of mustard (Table 2), expressing antisense *ACO*, was greatly enhanced compared to wild-type, as reported by Pua and Lee [125]. Shoot regeneration (%) in these transgenic lines was similar (about 90%, Table 1) to that reported when a chemical approach using silver nitrate and AVG was used [108,109]. In lemon, ACC and ETH treatments seems to decrease shoot regeneration in adult nodal segments, while silver thiosulfate treatments enhanced it [116]. Induced organogenesis in leaves of two genotypes of tomato (*Solanum pennellii* and F1: *Solanum pennellii* vs *Solanum lycopersicum* cv. Anl27; Table 1) revealed that an excessive amount of ETH or ACC negatively affected the percentage of explants showing buds in both genotypes, whereas lower concentrations of both ethylene precursors had no relevant effect [123]. Silver nitrate also negatively affected the number of explants with buds in *Solanum pennellii* and, interestingly, completely inhibited regeneration from F1 explants. Cobalt chloride decreased this percentage from F1 explants at only 21 µM, but it did not affect *Solanum pennellii*. ETH treatments showed a decrease regarding explants with shoots (%) for both genotypes similarly to silver nitrate treatments in *Solanum pennellii.* Although the negative effect of both ETH and silver nitrate in the number of shoots per explant with shoots (%) in *Solanum pennellii*, ACC treatments enhanced this parameter. These results show that different genotypes have different responses and sensitivity to ethylene and/or its modulation, affecting its regeneration capacity. 

In poplar (Table 1), ethylene precursors ACC (5 µM) and ETH (10 µM) positively affect regeneration and subsequent plant development and growth, in parameters such as shoot elongation, induction and development of buds, and also root formation per explant [118]. On the other hand, AVG treatments in a range of 10–15 µM negatively affected these parameters. Experiments carried out during shoot regeneration in cotyledons of different *Arabidopsis* mutants (Table 2) showed that shoot regeneration decreased in ethylene insensitive mutants *etr1-1* and *ein2-1*, whereas it increased in ethylene constitutive response mutants *ctr1-1* and *ctr1-12*, and also in ethylene overproducer mutant *eto1-1*, suggesting a positive role for ethylene on *Arabidopsis* organogenesis [126].

### 7.2. Somatic Embryogenesis

The first studies of the SE process in carrot (Table 1) showed that inhibition of ACO, using cobalt chloride, increased the number of somatic embryos formed from embryogenic cell suspensions [115]. Furthermore, ethylene measurements confirmed that the number of somatic embryos increased with the decrease of ethylene production. The negative effect of ethylene in carrot SE was further confirmed with ETH treatments (69.2 and 692 µM), in which the number of somatic embryos formed decreased [115]. In robusta coffee [119], the development of somatic embryos from leaf explants of two genotypes greatly increased in the presence of silver nitrate treatments (30 and 60 µM). In this species, high silver nitrate concentrations (150 and 300 µM) reduced somatic embryo formation, perhaps, due to its toxicity to plant tissues, rather than due to an inhibitory effect on ethylene perception [119]. Interestingly, the two different genotypes showed a similar great yield in somatic embryo formation (around +60%) but at different silver nitrate concentrations, one at 30 and the other at 60 µM, suggesting different sensitivities to ethylene according to the genotype. The effect of ethylene modulators on somatic embryo development from embryogenic calli in robusta coffee was also analyzed [120]. The data showed that somatic embryo development was poor in the control (with no modulators used), whereas in the presence of silver nitrate, cobalt chloride, or salicylic acid, this recalcitrance seemed to be reversed and an increase in somatic embryo formation was observed. However, results showing a positive effect of ethylene on direct SE from leaf explants of the same species were also reported [127]. SE was directly induced in leaf explants by cytokinin in both the Fuentes et al. [119] and Hatanaka et al. [127] studies. However, a preculture of 5 days of leaf explants in induction medium without silver nitrate was carried out in the work of Fuentes et al. [119]. This can somehow explain the differences observed, suggesting that ethylene can be necessary in an initial phase to promote embryogenic competence in leaf explants, but later disrupt somatic embryo development [119]. This could justify why an initial ethylene inhibition disrupted direct SE from leaf explants [127], but its later inhibition did not affect somatic embryo development [119].

Preliminary studies carried out in our lab with a solanaceous tree species (*Solanum betaceum*; commonly known as tamarillo), showed that at the induction stage, the leaf explants exposed to AgNO3 and AVG only produced non-embryogenic calli, whereas ETH significantly increased the induction of embryogenic tissue [128]. Furthermore, ETH treatments accelerated the induction of embryogenic calli. During somatic embryo development, following embryogenic callus transfer to an auxin-free medium, the treatment with AgNO3 and AVG enhanced the number of somatic embryos developed from embryogenic calli, whereas the presence of ETH blocked development beyond the globular stage. Moreover, as found in other species, ethylene may be involved at different steps of SE induction of tamarillo, from induction to embryo development. These results indicate that ethylene certainly has a role during somatic embryo formation and development in *S. betaceum*, but further studies are necessary to clarify how this hormone interacts with other players that are also crucial for somatic embryo formation in this species, such as auxins and high sucrose levels [93]. 

In black spruce (Table 1), a gymnosperm, the effect of inhibitors of ethylene biosynthesis and perception, AOA and silver nitrate, and also of ACC and pure ethylene on somatic embryo maturation from two cell lines with different embryogenic capacity was analyzed [114]. Inhibiting ACS activity with 10 µM AOA increased the total number of somatic embryos developed from a cell line with low embryogenic capacity. However, with the same AOA treatment, a decrease in the number of somatic embryos developed from a high embryogenic cell line was observed. Similar results were reported for both lines when treated with silver nitrate. Adding pure exogenous ethylene (1069 µM) along with 10 µM AOA treatments reverted the previous effect of AOA observed in both cell lines, confirming that somatic embryo formation in this species is somehow regulated by ethylene. Nevertheless, higher AOA concentration (100 µM) has a negative impact in somatic embryo formation on both lines. When using the ethylene precursor ACC, the formation of somatic embryos decreased at 100 µM for both cell lines, and 10 µM also negatively affected this parameter, although only for the high embryogenic line. Exogenous treatments with pure ethylene at 1069 µM also decreased the number of somatic embryos for the low embryogenic line, without affecting significantly the line with high embryogenic capacity. The differences in embryogenic capacity seem to be related to differences in ethylene production in both cell lines. While in high embryogenic cell lines, ethylene production remains constant and low, at a supposedly optimal concentration low embryogenic cell lines have a higher ethylene production, showing a supposedly supraoptimal ethylene concentration [114]. Therefore, in this species, ethylene below or above an optimal concentration seems to be directly correlated with low embryogenic regeneration capacity.

In Scots pine [133], the expression of *ACS1* and *ACS2* genes during different SE stages in cell lines with different embryogenic capacity was studied (Table 2). Although the *ACS1* gene was similarly expressed throughout the different lines and in different SE developmental stages, the *ACS2* gene was markedly expressed only in somatic embryos, along with high ethylene production. The line with the higher embryogenic capacity showed a positive correlation between the number of somatic embryos developed, ethylene production and *ACS2* expression. Ethylene production increased throughout somatic embryo development, was greatly detected at the cotyledonary stage, and decreased during subsequent somatic embryo conversion and germination [133]. 

In spinach (Table 1), SE was induced from root explants using inhibitors of ethylene biosynthesis and signaling—silver nitrate and AVG—and also ETH. A decreased induction of embryogenic calli from spinach roots was observed in silver nitrate and AVG treatments, while using ETH, at 10 and 100 µM, significantly enhanced the percentage of embryogenic calli induced [121]. However, during somatic embryo formation, ETH negatively affected the number of somatic embryos produced per explant, whereas inhibiting ethylene perception with silver nitrate greatly increased the number of somatic embryos developed per callus, the highest being a threefold increase (at 10 µM) when compared with the control. Once again, the results show that the effect of ethylene depends on the stage of somatic embryo differentiation. 

Inhibition of ethylene perception in summer snowflake (Table 1) by silver nitrate or using potassium permanganate, as an ethylene absorbent, increased the proliferation of embryogenic calli [122]. However, this proliferation was negatively affected by silver thiosulfate. Despite the negative impact of ACC on the embryogenic calli proliferation, an increase in the number of somatic embryos formed and their posterior maturation was reported in ACC treatment. A great number of somatic embryos at torpedo stage in the presence of ACC was observed, when compared with both the control and other treatments, suggesting that ethylene is essential for somatic embryo development in *Leucojum aestivum*. 

Assays in *Arabidopsis* (Table 2) to evaluate the effect of ACC and ethylene mutants in somatic embryo formation from embryogenic calli [129] showed that as ACC concentration rose, the number of somatic embryos formed per embryogenic callus decreased. A similar effect was observed in both the ethylene overproducer mutant, *eto1-1*, and the ethylene constitutive response mutant, *ctr1-1*. Along these results, embryogenic calli treated with ACC (200 µM) or from the mutants *eto1-1* and *ctr1-1* showed a downregulation in *YUCCA* genes expression. *YUCCA* genes are known to encode for the key enzymes of auxin biosynthesis [134] and their requirement to induce SE in *Arabidopsis* from embryogenic calli is known [129]. Furthermore, ACC (200 µM) and the mutant *eto1-1* showed a disruption in local auxin biosynthesis and consequent distribution [124]. In fact, the quadruple mutants *yuc1*, *yuc4*, *yuc10*, and *yuc11*, which impair local auxin distribution, also showed a reduction in the number of somatic embryos formed from embryogenic calli, similar to what was observed in ACC treatments at 200 µM. Taken together, these results provided evidence of the hormonal regulation required to induce somatic embryos in Arabidopsis’ embryogenic tissue, suggesting that ethylene negatively affects SE in this species through inhibition of auxin biosynthesis and its local distribution [129]. Moreover, somatic embryo initiation in auxin-rich medium and in auxin-free medium were compared [129]. Exogenous auxin stimulated ethylene production, and its removal from the medium enhanced somatic embryo initiation along with lower levels of ethylene. Using immature zygotic embryos as explants [130], at first glance, the ethylene effect is not so clear when only ethylene modulators are used, since both ACC and silver nitrate negatively affected the SE process. In other words, SE efficiency (i.e., the percentage of explants that formed embryos) and SE productivity (i.e., the average of somatic embryos produced per explant) decreased in the presence of ETH, but also decreased in the presence of inhibitors of ethylene biosynthesis and perception (see Table 2 for more details). However, AVG and cobalt chloride at low concentration (1 µM) do not affect these parameters. Using ethylene mutants, with different and opposite phenotypes, a decrease in the SE efficiency and productivity was also observed, similarly to what was observed with the chemical modulation. Taking these results into account, and knowing that in control conditions SE efficiency reaches 90%, it can be concluded that immature zygotic embryos have *per se* an optimal ethylene production at the induction conditions, which includes 5 µM 2,4-D and 20 g.L^−1^ sucrose [130], promoting an efficient SE process. Furthermore, in this work, the authors also studied the effect of the *ETHYLENE RESPONSE FACTOR022* (*ERF022*) gene on *Arabidopsis* SE. *ERF022* is a member of the ethylene response factor (ERF) family genes, which are associated with plant responses to stress [135]. It was also found that a strong inhibition of the *ERF022* gene is associated with SE induction in *Arabidopsis* [130]. However, its knock-out, in the *erf022* mutant, exhibited a reduced capacity for SE [136]. Confronting these results, it was suggested that *ERF022* expression is required to induce SE in *Arabidopsis*. In this regard, the authors [130] also studied a putative molecular function behind this phenomenon. They reported a downregulation of *ACS7*, *ERF1*, and *ETR1* gene expression, suggesting that *ERF022* negatively controls ethylene biosynthesis and perception. Furthermore, and even more interestingly, they found that the *erf022* mutant exhibited a great inhibition of *LEAFY COTYLEDON2* (*LEC2*) gene expression, along with the impaired capacity of SE induction. *LEC2* was reported as a promoter of SE in *Arabidopsis* via *YUCCA*-mediated auxin biosynthesis [137]. In line with this, the *erf022* mutation was correlated with a downregulation of *YUC1* and *YUC4* gene expression, along with reduced levels of the endogenous auxin, IAA. This study provides evidence that SE induction from immature zygotic embryos in *Arabidopsis* is based on an ethylene–auxin crosstalk, mediated by *ERF022*–*LEC2* interaction [130]. 

In the case of alfalfa, a model often used to analyze SE, the embryogenic competence, in the presence of NBD, an inhibitor of ethylene perception, was considerably reduced [110]. Further embryo maturation also decreased in the presence of this ethylene receptor’s competitive inhibitor. In another study with this species [111], both embryogenic calli proliferation and somatic embryo maturation decreased in the presence of 50 µM AVG. The same group [112] also studied the effect of the inhibitors of ethylene biosynthesis and perception on SE induction in alfalfa (see in detail in Table 1). In agreement with the two last studies, these modulators negatively affected some important SE stages, such as somatic embryo formation, further development, and maturation. AVG and salicylic acid showed a great disruption on somatic embryo development, with a great decrease in the number of somatic embryos in later developmental stages, such as at the cotyledonary stage; in some cases with only globular somatic embryos being observed. Interestingly, in *Medicago truncatula* (Table 2), a species from the same genus as alfalfa (*Medicago sativa*), treatments with the ethylene precursors ACC and MGBG increased the number of somatic embryos developed per embryogenic callus, while silver nitrate and AVG greatly decreased the somatic embryo formation [131]. Furthermore, when high silver nitrate and AVG concentrations (100 µM) were tested, somatic embryo formation was completely inhibited. Moreover, a positive correlation between *ACS* and *ACO* gene expression and the genotype with higher embryo production capacity was found, suggesting that ethylene is required for somatic embryo formation and development in this species [131]. Interestingly, *ACS* and *ACO* expression in somatic embryos were found to be similar to the patterns of expression detected in zygotic embryos. The authors also reported an upregulation of ethylene responsive genes when a transcriptional profile analysis of embryogenic calli was carried out. It was found that a member of the ERF family, denominated *MtSERF1*, was highly expressed in embryogenic calli and strongly expressed in globular somatic embryos. The same gene was weakly associated with low embryogenic capacity. Furthermore, ethylene-dependent *MtSERF1* expression was proven by its inhibition with AVG and silver nitrate. These data, together with the fact that silencing *MtSERF1* using RNAi completely inhibited the somatic embryo formation [131], proving that ethylene is a key factor during SE induction—probably mediating the action of other hormones, such as auxins and cytokinins. *MtSERF1* orthologs genes [113] were found in *Arabidopsis thaliana* (*At5g61590*) and soybean (*GmSERF1* and *GmSERF2*), which are involved in the regulation of *AGAMOUS-Like15* (*AGL15*). This gene promotes SE in the SAM of *Arabidopsis* seedlings [138] and in soybean [139], being upregulated by auxin, in particular, 2,4-D [140]. Ethylene modulation has been tested in both soybean and *Arabidopsis* SE (see Table 1). The *Arabidopsis* SAM-SE system [124], is highly promoted by the ethylene precursor ACC, with an increase in number of seedlings with somatic embryos. Furthermore, a reduction in the number of seedlings with embryos was observed in the presence of the inhibitors of ethylene biosynthesis and perception, such as AVG, cobalt chloride, and silver nitrate, suggesting a positive correlation between SE and ethylene. Similar results were observed in different cultivars of soybean, with different embryogenic capacity [113]. The number of somatic embryos increased with ACC treatments and decreased with AVG, with the positive impact of ACC treatments highly marked in the recalcitrant cultivars. The relationships between ethylene/*At5g61590*/*AGL15* and ethylene/*GmSERF1*/*GmAGL15* in the induction of SE of *Arabidopsis* and soybean, respectively, were confirmed by the transcript accumulation of *At5g61590* and *GmSERF1* in response to ethylene modulators. Both *At5g61590* and *GmSERF1* transcript levels were enhanced in ACC treatment, and, in turn, decreased transcript levels of both genes in the presence of inhibitors of ethylene biosynthesis and perception were reported. Transcription levels of *AGL15* were also upregulated or downregulated in the presence of ACC or AVG, respectively. Thus, it can be concluded that ethylene regulates SE induced by 2,4-D in *Arabidopsis* seedlings and in soybean cotyledons, by regulating the ortholog genes of *MtSERF1* and subsequent regulation of AGL15, based on an ethylene–auxin crosstalk. Note that *ERF022,*
*MtSERF1,*
*At5g61590*, and *GmSERF1* are all members of the ERF family. 

The effect of ethylene on SE seems also to be species-specific, and within the same species it can affect the regeneration process differently regarding each specific stage (Table 3). For instance, in alfalfa [111,112], ethylene has a stimulatory effect independently of the SE stage, while in spinach [121] ethylene promotes embryogenic calli induction but disrupts somatic embryo development. 

Initial explant and medium compositions are also critical factors to achieve SE. In *Arabidopsis* seedlings and soybean cotyledons ethylene had a stimulatory effect in somatic embryo formation in the presence of auxins [113], while in an auxin-free medium ethylene disrupted the somatic embryo formation from *Arabidopsis*’ embryogenic calli [129]. Furthermore, in *Mendicago truncatula* [131] ethylene is essential to promote somatic embryo formation from embryogenic calli in the presence of auxins and cytokinins. Auxins and cytokinins act as stress inducers, leading to SE initiation and callus formation [3]. The role of ethylene in SE in some species seems to be related to stress mediated by ethylene in response to auxin and cytokinins. A possible molecular framework is proposed (Figure 3). Initial stress stimulus by auxins, such as 2,4-D, leads to an increase in ethylene production [130]. An initial increase in ethylene biosynthesis seems to be required for the induction of specific ERF genes [130] essential to induce SE [136]. ERFs stimulate LEC expression [130] and consequent YUC expression, needed for SE induction. Initial higher levels of ethylene (induced by auxins) seem to downregulate YUC levels [129]; however, ethylene levels may tend to decrease over time, as stress levels decrease, leading to a lesser inhibition of YUC. Somatic embryo development can be induced in an auxin-free medium [129], but also in the presence of auxins [113] or auxins plus cytokinins [131]. Based on the studies of Bai et al. [129], in *Arabidopsis*, an auxin-free medium leads to a downregulation in ethylene levels, and consequent upregulation of YUC levels needed to induce somatic embryo development may be in a stress-independent process. In some species, somatic embryo development can be induced in the presence of stress factors, such as auxins and cytokinins [113,131]. In this specific situation, somatic embryo development seems to be induced in a response to stress mediated by ethylene. Ethylene leads to an upregulation of SERF1 gene expression [113,131] with consequent upregulation of AGL15 levels [113]. AGL15 stimulates LEC expression [141] leading to somatic embryo development. Taking into consideration the role of the ERF family in response to stress, ethylene may affect SE differently based on stress-response signaling induced by the hormonal stress caused by auxin and/or cytokinin.

## 8. Conclusions and Future Prospects 

The compiled literature presented here in can give us a great insight into how ethylene affects regeneration processes. It is markedly noted that ethylene affects regeneration depending on the species, the explant, and the stress conditions. We know in advance that a successful regeneration is a requirement to the success of micropropagation processes. Thus, ethylene modulation emerges as imperative regarding the optimization of micropropagation protocols. Modulation of culture conditions such as medium composition and culture atmosphere are some examples that can be further applied when the effect of ethylene on regeneration for each species is known. Moreover, the ethylene capacity in reverting recalcitrance in some species highlights the importance of ethylene modulation studies regarding in vitro regeneration. Further studies focused on the ERF members genes regulated by ethylene in response to stress-induced regeneration can contribute to unveil the mechanisms behind the highly regeneration capacity observed in some genotypes. It may also explain why some genotypes and species are recalcitrant to regeneration. Ethylene seems to affect in vitro regeneration by stress-response signaling, with evident hormonal crosstalk, at least in some species. Considering all the literature, this hormone seems to be an important link between stress, regeneration, and development. 

## Figures and Tables

**Figure 1 plants-10-01208-f001:**
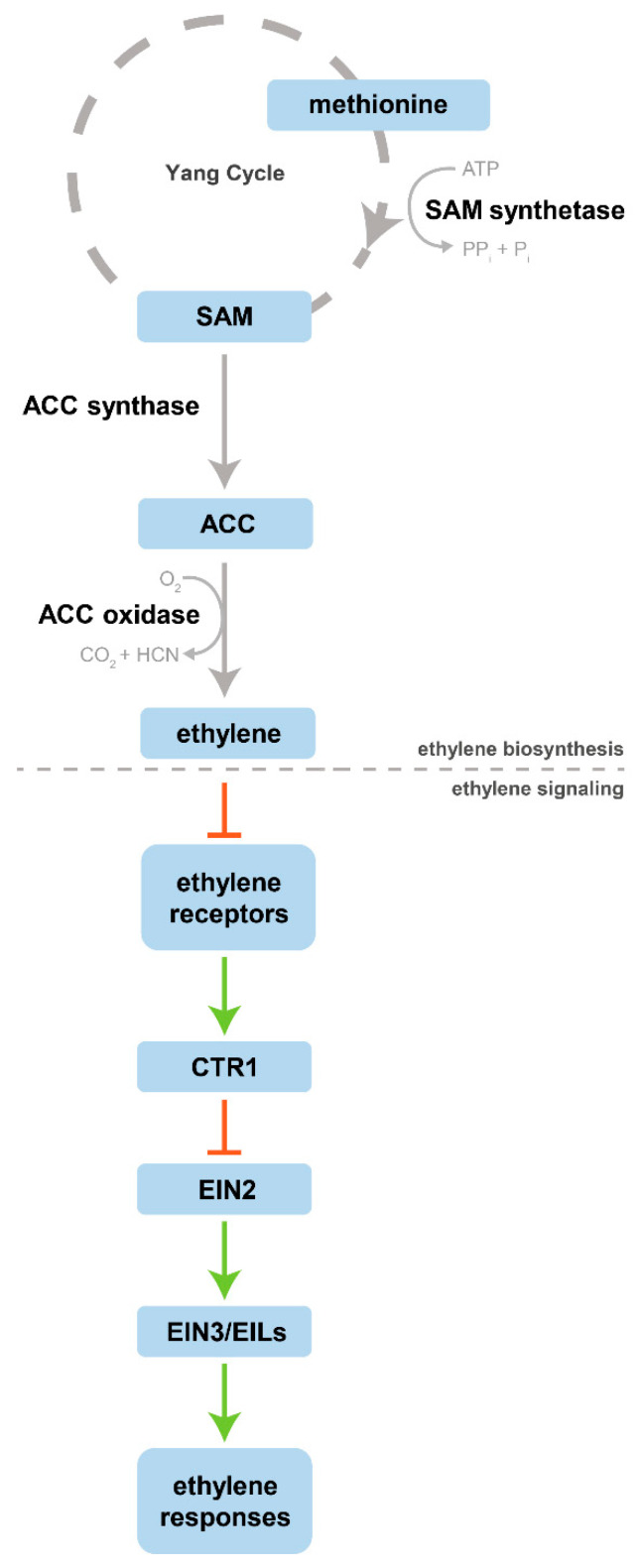
Schematic overview of ethylene biosynthesis and the canonical signaling pathway. Ethylene acts as an inverse agonist, inhibiting its receptors. This inhibition leads to a reduction in CTR1 activity, which allows the set of reactions downstream to occur, culminating in ethylene responses. The different pathways are separated by a dashed line. Inhibition steps are marked with an inhibitory arrow (red). Based on the signaling models of Wang et al. [10] and Binder [15].

**Figure 2 plants-10-01208-f002:**
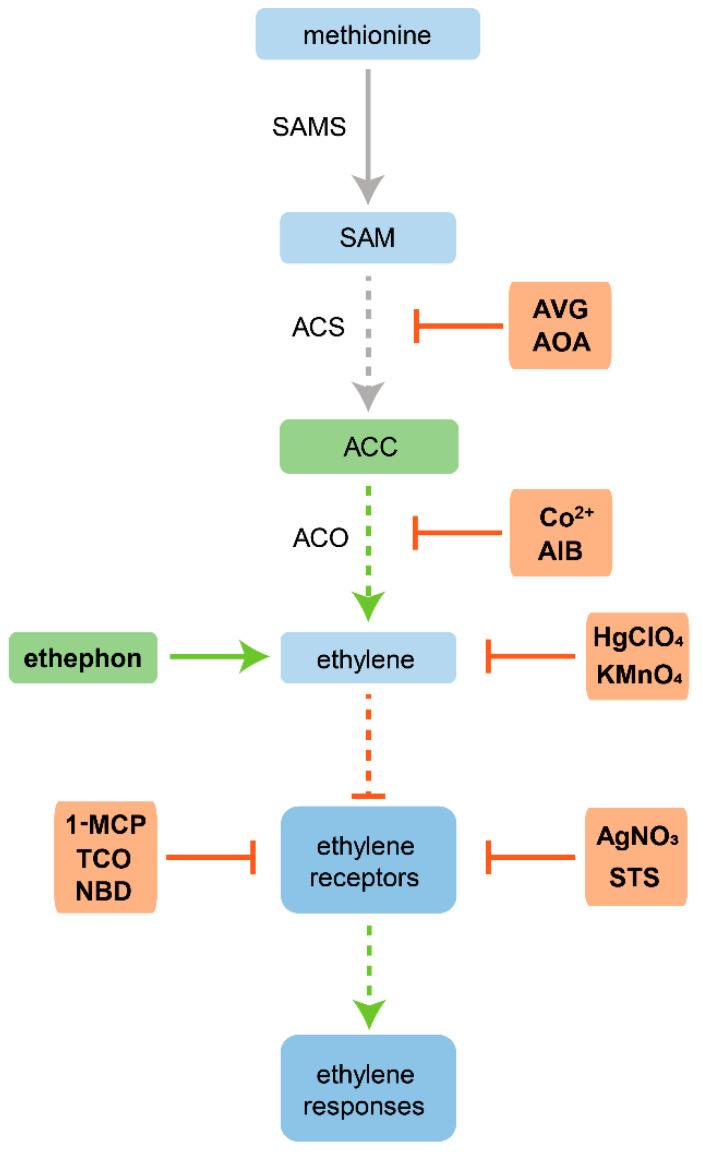
Schematic diagram of some of the principal ethylene modulators and their points of action. Ethylene precursors are shown in green followed by a green arrow. Inhibitors of ethylene biosynthesis and action are shown in orange followed by an inhibitory arrow (red). Dashed arrows indicate the pathway steps that can be affected by modulation. Based on Schaller and Binder [22].

**Figure 3 plants-10-01208-f003:**
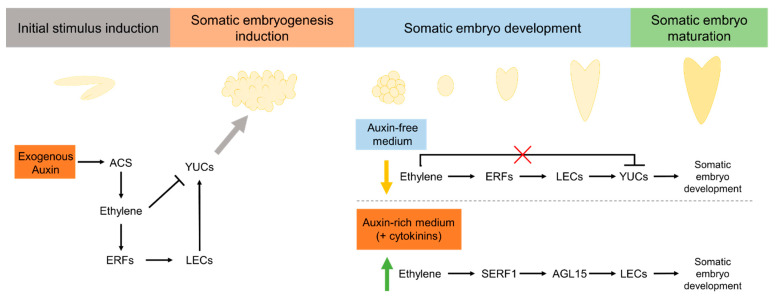
Possible molecular framework for the effect of ethylene on SE process. Exogenous auxins enhance ethylene production in response to stress. Specific transcriptional factors from the ERF family are activated, leading to an increase in *LEC* and *YUC* expression levels and consequent SE induction—based on the molecular mechanisms proposed by Nowak et al. [130] in *Arabidopsis* and the studies of Ikeuchi et al. [3]. In an auxin-free medium, ethylene production decreases, leading to an increase in YUC levels and somatic embryo development—based on *Arabidopsis* studies [129]. In the presence of auxin, somatic embryo development depends on the specific ERFs to be induced, SERF1 and GL15, perhaps as a consequence of stress induced by auxins and cytokinins—based on studies in *Mendicago truncatula* [131] and soybean [113].

**Table 1 plants-10-01208-t001:** Effect of ethylene modulators on different regeneration systems of diverse plant species. Ethylene modulators in the effect column showed an increase (↑) or a decrease (↓) in the respective parameter when compared to control.

Plant Species	Process	Explant	Modulation	Effect	Ref.
**Alfalfa** (*Medicago sativa*)	SE	Petioles and petiole-derived EC	Between 0 and 410 µM NBD	Callus induction/explant and embryo maturation: ↓ as NBD concentration rises, but the calli preinduced with NBD form somatic embryos	[110]
			5 and 50 µM AVG	Somatic embryo differentiation and EC proliferation: ↓ in 50 µM AVG; Somatic embryos at the cotyledonary stage are reduced in all treatments, but EC induction is not significantly affected by AVG	[111]
		EC induced from petioles (suspension culture in liquid medium)	1, 10, 100, or 500 µM AVG or SA; 0.045, 0.09, or 0.112 µM 1-MCP	EC proliferation: ↓ in 10, 100 and 500 µM SA (−50, −70 and −90%, respectively); ↓ in all AVG treatments (−25, −40, −55 and −90%); ↓ in all 1-MCP treatments (−30, −40 and −60%) Embryogenic potential of treated suspension cultures (number of somatic embryos formed): ↓ in all SA treatments (−25, −70, −80 and −90%); ↓ in 10, 100 and 500 µM AVG (−30, −50 and −90%, respectively) and also ↓ in all 1-MCP treatments Somatic embryo development: ↓ somatic embryos at the cotyledonary stage in 1 and 10 µM SA (−25 and −70%, respectively) but ↑ somatic embryos at the globular stage in 100 µM SA; ↑ globular embryos in all AVG treatments; cotyledonary embryos ↓ in 1, 10 and 100 µM AVG (−25, −35 and −40%, respectively); in all 1-MCP treatments ↑ number of globular embryos but further development was blocked	[112]
***Arabidopsis thaliana***	SAM-SE system ^a^	Seedlings ^a^	25 µM ACC; 10 µM AVG or AgNO_3_; 100 µM CoCl_2_	Seedlings with embryos (%): ↑ in ACC treatment (around 40%); ↓ in AVG, AgNO_3_ and CoCl_2_ treatments (<10%); control: around 25%	[113]
**Black spruce** (*Picea mariana*)	Somatic embryo maturation	Two embryogenic cell lines, with low (a) or high (b) embryogenic capacity	5, 10, and 100 µM ACC; 0.5, 1, and 2 mM AgNO_3_; 5, 10, and 100 µM AOA; 178, 356, and 1069 µM C_2_H_4_	Total embryos formed: ↓ in 100 µM ACC for (a) and ↓ in 10 and 100 µM ACC for (b); ↓ in 1069 µM C_2_H_4_ for (a), but C_2_H_4_ treatments not affect (b) significantly; ↑ in 10 µM AOA and ↓ 100 µM AOA for (a); ↓ in 10 and 100 µM AOA for (b); ↑ in 1 mM AgNO_3_ for (a) and ↓ in 1 and 2 mM AgNO_3_ for (b) Ethylene production in both lines: C_2_H_4_ production increased during somatic embryo maturation for (a) and C_2_H_4_ production was maintained constant and low for (b)	[114]
**Carrot** (*Daucus carota*)	SE	Embryogenic cell suspension induced from hypocotyls	10, 20, 50, and 100 µM CoC1_2_; 69.2 and 692 µM ETH	Number somatic embryos formed (No./mL of cell suspension): ↑ in 10, 20 and 50 µM CoC1_2_, but ↓ in 100 µM (best treatment, 79 No./mL at 50 µM; control: 23 No./mL); ↓ in both ETH treatments; ↓ in 50 µM CoC1_2_ + 69.2 or 692 µM ETH	[115]
**Gloxinia** (*Sinningia speciosa*)	Org ^b^	Leaf explants	6.24, 31.22, 62.43, and 124.87 µM AVG; 7.7, 38.5, 77, and 154 µM CoCl_2_; 3, 15.25, 30.5, and 61 µM STS	Regeneration (%) and shoots per explant: ↑ in 15.25 and 30.5 µM STS (best treatment, 15.25 µM STS, 40% more shoots/explant); ↑ in 6.24 µM AVG or 7.7 µM CoCl_2_ and ↓ in 62.43 and 124.87 µM AVG or 77 and 154 µM CoCl_2_	[105]
**Lemon** (*Citrus limon*)	Org	Adult nodal segments from two cultivars (a) Verna 51 and (b) Fino 49	10, 20, and 30 µM ACC, CoCl_2_ or STS; 5, 10, and 20 µM ETH	Regeneration (%): ↓ in 10, 20 and 30 µM ACC for both; ↓ in 10 and 20 µM ETH for (b); ↑ in 10 and 20 µM STS for both; ↓ in 10 and 20 µM CoCl_2_ for (b) and ↓ 30 µM CoCl_2_ for both Buds per explant: ↓ in 10, 20 and 30 µM ACC for both and ↓ in 30 µM CoCl_2_ for (a)	[116]
**Yellow passionfruit** (*Passiflora eduli*s)	Axillary bud culture	Nodal segments	3 and 10 µM ACC, STS, or AVG	Buds and leaf area per explant: ↓ in 3 and 10 µM ACC and ↑ in 3 and 10 µM STS or AVG treatments Shoot length per explant: ↓ in 3 and 10 µM ACC treatments	[117]
**Melon** (*Cucumis melo*)	Org	Cotyledons ^b^	60 or 120 µM AgNO_3_; 69.2 or 138.4 µM ETH	Shoot regeneration (%): ↑ in all AgNO_3_ treatment for all genotypes; line with best shoot regeneration, 75% at 60 µM and 68% at 120 µM (control 35%) Shoot regeneration (%) for the best line: ↓ in both ETH treatments (19% and 10% at 69.2 µM and 138.4 µM ETH, respectively)	[106]
**Mustard** (*Brassica juncea* L.)	Org	Hypocotyls	17.66 µM AgNO_3_	Shoot regeneration (%): ↑ in AgNO_3_ treatment with 95.89% shoot regeneration (control: 14.6%)	[108]
		Leaf disc and petioles	20 µM AgNO_3_ and 5 µM AVG; AgNO_3_ or AVG with 10, 25, or 50 µM ETH (combined)	Shoot regeneration (%) from both explants: ↑ in both AgNO_3_ and AVG treatments, with 80–90% (control: 20–30%) Shoot regeneration (%) in combined treatments (leaf explants): ↓ in 25 or 50 µM ETH + AVG (50 µM ETH + AVG almost inhibited regeneration and slight ↓ in ETH + AgNO_3_)	[109]
	Plant growth	Shoots	20 µM AgNO_3_, 5 µM AVG, or 50 µM ETH	Plant growth parameters, such as plant height, number of leaves, number of roots and root length: ↓ in both AgNO_3_ and ETH treatments (AVG does not have a very significant effect on the same parameters)	
**Poplar** (*Populus tremula*)	Org	Nodal segments	5, 10, and 15 µM AVG; 0.5, 1 and 5 µM ACC; 5 and 10 µM ETH	Shoot elongation and number of buds and roots/explant: ↓ in 10–15 µM AVG, ↑ in 5 µM ACC and ↑ in 10 µM ETH	[118]
**Robusta coffee** (*Coffea canephora*)	SE	Leaf squares (two genotypes)	30, 60, 150, and 300 µM AgNO_3_	Number of embryos per explant: ↑ in 30 and 60 µM treatment and ↓ in 150 and 300 µM treatment; One genotype shows the greatest yield at 30 µM (+57%) and the other at 60 µM (+60%)	[119]
		EC developed from hypocotyl and leaf explants	20 and 40 µM AgNO_3_, CoCl_2_, or SA	Calli responded for embryogenesis (%): ↑ in all AgNO_3_ treatments (best treatment, 40 µM, 48%); ↑ in all CoCl_2_ treatments (best treatment, 40 µM, 28%); ↑ in all SA treatments (best treatment, 40 µM, 32%), control 5% Number of somatic embryos per callus: ↑ in both all AgNO_3_ and AVG treatments (best treatment, at 40 µM, 153 and 45 embryos, respectively); in all SA treatments, only pro-embryogenic nodular masses appeared (control did not produce somatic embryos)	[120]
**Soybean** (*Glycine max*)	SE	Cotyledons from cultivars with different embryogenic capacity	10 µM ACC or AVG	Somatic embryo production: ↑ in ACC treatment for two recalcitrant cultivars (slight increase, but not significantly, for cultivar with high embryogenic capacity); ↓ in AVG treatment (almost inhibited) for both two recalcitrant cultivars and cultivar with high embryogenic capacity	[113]
**Spinach** (*Spinacia oleracea*)	SE	Roots	1, 10, and 100 µM ETH or AgNO_3_ 1 and 10 µM AVG	Embryogenic callus (%): ↓ in all AgNO_3_ and AVG treatments and ↑ in 10 and 100 µM ETH (only in combination w/ 0.1 µM GA_3_) Calli forming embryos (%): ↓ in all ETH treatments (somatic embryos formation inhibited at 100 µM) and ↑ in all AgNO_3_ treatments Number of embryos/calli: ↓ in all ETH treatments and ↑ in all AgNO_3_ treatments (best treatment, 10 µM AgNO_3_, 3× more embryos)	[121]
**Summer snowflake** (*Leucojum aestivum*)	SE	EC	10 µM ACC, AgNO_3_, or STS KMnO_4_ (4.5 g solid)	EC increment (%):↑ in both AgNO_3_ and in KMnO_4_; ↓ in both ACC and STS treatments Somatic embryo induction and maturation: ↑ in ACC treatment Length of plantlets development: ↑ in KMnO_4_ treatment	[122]
**Tomato** (*Solanum pennellii* and *Solanum lycopersium*)	Org	Leaves from two genotypes (*Solanum pennellii* and F1: *Solanum pennellii* vs *Solanum lycopersicum* cv. Anl27)	5.8, 14.5, 29, and 58 µM AgNO_3_; 4.2, 10.5, 21, and 42 µM CoCl_2_; 9.8, 24.5, 49, and 98 µM ACC; 6.9, 17.2, 34.5, and 69 µM ETH	Explants with buds (%): ↓ in 14.5, 29 and 58 µM AgNO_3_ for *S. pennellii*; ↓ in 21 µM CoCl_2_ for F1; ↓ in 98 µM ACC for both genotypes; ↓ in 17.2, 34.5, and 69 µM ETH for *S. pennellii* and ↓ in 34.5 and 69 µM ETH for F1; (lowest percentage at 69 µM ETH for both *S. pennellii* and F1, with 20% and 16%, respectively) Explants with shoots (%): ↓ in 14.5, 29 and 58 µM AgNO_3_ for *S. pennellii*; ↓ in 21 µM CoCl_2_ for F1; ↓ in 24.5 and 98 µM ACC for both *S. pennellii* and F1; ↓ in all ETH treatments for both genotypes (lowest percentage at 69 µM ETH for both *S. pennellii* and F1, with 12% and 8%, respectively) Nunber of shoots per explant with shoots: ↓ in all AgNO_3_ treatments for *S. pennellii*; ↑ in 9.8, 49 and 98 µM ACC for *S. pennellii* (↑ ×2 more compared to control (around 5 shoots), with around 10 shoots); ↓ in 17.2, 34.5 and 69 µM ETH for *S. pennellii*; (lowest number at 58 µM AgNO_3_ and 69 µM ETH with 0.96 and 1, respectively) All AgNO_3_ treatments inhibited regeneration for F1	[123]

^a^ Shoot apical meristem-somatic embryo (SAM-SE) system, characterized by somatic embryos developed at the shoot apical meristem from seeds germinated in the presence of the synthetic auxin 2,4-dichlorophenoxyacetic acid (2,4-D) as described by Mordhorst et al. [124]. ^b^ Org—Organogenesis.

**Table 2 plants-10-01208-t002:** Effect of ethylene modulators, mutants, or transgenic lines on different regeneration systems of diverse plant species. Ethylene modulators/mutants or transgenic lines in the effect column showed an increase (↑) or a decrease (↓) in the respective parameter when compared to control.

Plant Species	Process	Type of Explant	Modulation	Effect	Ref.
***Arabidopsis thaliana***	Org ^a^	Cotyledons	Ethylene mutants	Shoot regeneration (%): ↓ in ethylene insensitive mutants (*etr1-1* and *ein2-1*), ↑ in ethylene constitutive response mutants (*ctr1-1* and *ctr1-12*) and ↑ in ethylene overproduction mutant (*eto1-1*)	[126]
	SE	Embryonic calli (induced from primary somatic embryos preinduced from immature zygotic embryos)	10, 20, 50, 100, 150, and 200 µM ACC; ethylene mutants	Somatic embryo regeneration/embryonic calli: ↓ as ACC treatment concentration rises, 100 and 150 μM greatly decreases somatic embryo production and 200 μM almost inhibited its regeneration; ↓ in both ethylene overproduction mutant (*eto1-1*) and ethylene constitutive response mutant (*ctr1-1*); *ctr1-1* almost inhibited somatic embryo formation; (ethylene insensitive mutants (*etr1-3* and *ein2-1*) and double *ACS* mutant (*acs2-1 acs6-1*) do not affect somatic embryos production)	[129]
		Immature zygotic embryos	1, 5, 10 µM ACC; 1, 10 µM CoCl_2_; 1, 10, 15 µM AVG; 1, 10, 100 µM AgNO_3_ or 250 mM KMnO_4_; ethylene mutants	Explants that formed somatic embryos (%): ↓ in 1, 5, and 10 µM ACC; in 10 µM CoCl_2_; in 10 and 15 µM AVG; in 10 and 100 µM AgNO_3_ and also in 250 mM KMnO_4_ treatments (lower % at both 10 µM ACC and 10 µM CoCl_2_ treatments around 20%, control around 90%) Average number of somatic embryos produced/explant: ↓ in 1, 5, and 10 µM ACC; in 10 µM CoCl_2_; in 10 and 15 µM AVG and also in 100 µM AgNO_3_ treatments Both parameters ↓ in ethylene insensitive mutants, in ethylene constitutive response mutants, and in ethylene over- and under-producer mutants	[130]
***Medicago truncatula***	SE	Two different genotype leaf-derived EC lines, with different embryo production capability	0.1, 1, 10, and 100 µM ACC, MGBG, AgNO_3,_ or AVG; *ACS* and *ACO* expression; Mt*SERF1* ^b^ knockdown (using RNAi)	Number of somatic embryos developed/explant: ↑ in 1 and 10 µM ACC and in 10 and 100 µM MGBG, best treatments 10 µM ACC and 100 µM MGBG, = around 35 embryos/explant (control = around 12); ↓ in 1 and 10 µM AVG or AgNO_3_ and completely inhibited at both 100 µM AVG and AgNO_3_ treatments; *ACS and ACO expression*: ↑ in line with highly embryo production capacity; *MtSERF1* knockdown: Disrupt somatic embryo production	[131]
**Melon** (*Cucumis melo*)	Org	Leaves and cotyledons	Transgenic plant line expressing antisense *ACO*; 50 or 100 µM ETH	Shoot regeneration (%) from leaf explants: ↑ in transgenic line with 53% (control 15%) Shoot regeneration (%) from cotyledon explants: ↑ in transgenic line with 37% (control 13%) Shoot regeneration (%) from both explants in response to ETH: ↓ in all ETH treatments (for both explants); transgenic leaf explants + 50 µM ETH treatment shows 5% shoot regeneration and + 100 µM ETH treatment regeneration was inhibited; transgenic cotyledon explants + 50 µM ETH treatment shows 8% shoot regeneration and + 100 µM ETH treatment shows 1%	[107]
**Mustard** *(Brassica juncea)*	Org	Leaf discs and Hypocotyl segments	10 transgenic plant lines expressing antisense *ACO*; 5 µM AVG, 10 µM AgNO_3_, or 50 µM ETH (alone or combined)	Shoot regeneration (%) from leaf explants: ↑ in 9 transgenic lines, between 58% and 92% (control 12–16%), 4 best lines % (83, 79, 80, 92); Shoot regeneration (%) from leaf explants in response to inhibitors and/or ETH: ↑ in WT plants + AVG or + AgNO_3_ treatments (similar % compared with the 4 transgenic lines w/o treatment), (WT plants + AgNO_3_ + ETH = around 60%); Both WT plants + AVG + ETH (%) and the 4 transgenic lines + ETH (%) are similar to control; Shoot regeneration (%) from hypocotyl explants: ↑ in the 4 best transgenic lines, 85–95% (control = around 5%)	[125]
**Potato** (*Solanum tuberosum*)	Plant growth; Callus formation	Nodal segments with unfolded leaf	Different ventilations: (a) sealed with silicone rubber bungs; (b) capped with polypropylene discs; and (c) forced ventilation; 3 µM AgNO_3_ or 2 µM ACC, applied to cultures under different ventilations	Leaf area/explant: ↑ in both (a) + AgNO_3_ and (b) + AgNO_3_; ↑ in (a) + ACC and ↓ in (b) + ACC Steam length: ↑ (a) + ACC and ↓ in both (b) + ACC and (c) + ACC Roots/explant: ↑ (a) + ACC and ↓ in both (a) + AgNO_3_ and (b) + AgNO_3_ Root length/explant: ↓ in both (a) + ACC and (b) + ACC and ↑ in both a) + AgNO_3_ and (b) + AgNO_3_ Ethylene concentration in vessels: low ethylene concentration in both (b) and (c) vessels (similar) and high concentration in (a) vessels	[132]
**Scots pine** **(*Pinus sylvestris*)**	SE	Embryogenic cell line cultures with distinct embryogenetic capacity	*ACS1* and *ACS2* expression and ethylene production during different SE stages	*ACS1* transcript is accumulated throughout the lines with different embryogenic capacity and also in somatic embryos, similarly; *ACS2* transcript is accumulated only in somatic embryos (the ethylene production is only greatly detected in somatic embryos); ↑ *ACS2* gene expression levels in the cell line with higher embryogenic capacity; embryos at cotyledonary stage showed highly ethylene production and during germination into plantlets ethylene production is greatly reduced	[133]

^a^ Organogenesis. ^b^
*SOMATIC EMBRYO RELATED FACTOR1* (*SERF1*), the expression of which is dependent on ethylene biosynthesis and perception. [131].

**Table 3 plants-10-01208-t003:** Differences of ethylene effect depending on the stages of SE. Ethylene showed a stimulatory (↑) or an inhibitory (↓) effect in the respective stages regarding the use of different ethylene modulators.

Plant Species	Embryogenic Callus Induction	Embryogenic Callus Proliferation	Somatic Embryo Development	Somatic Embryo Maturation	References
Alfalfa	= ^a^	↑	↑	↑	[111,112]
*Arabidopsis thaliana*; Carrot; Robusta coffee	NA	NA	↓	NA	[129,115,119,120]
*Medicago truncatula; Soybean*	NA	NA	↑	NA	[131,113]
Scots pine	NA	NA	↑	↑	[133]
Spinach	↑	NA	↓	NA	[121]
Summer snowflake	NA	↓	↑	↑	[122]

^a^ Did not affect significantly. NA: Not available.

## Data Availability

Not applicable.

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
