# Peer review of "Modulation of Organogenesis and Somatic Embryogenesis by Ethylene: An Overview"

_plants, 2021, doi:10.3390/plants10061208_

Round 1

Reviewer 1 Report

The manuscript by Neves et al comprehensively reviews the literature on regeneration and the role of ethylene in this process in plants. They focus on this process and refer to other recent reviews on biosynthesis and signaling of ethylene.

I have the following suggestions:

1) On the SE process, apparently differences in the role of ethylene occur depending on the phase (Embryogenic callus induction, SE growth and further). Could the authors foresee a table or figure in which per phase the role of ethylene is indicated (stimulatory, inhibitory). This sequence of the ethylene effect per phase will depend on the plant species, but it seems some plant tissues have the same mode of ethylene effect sequence, and can thus be grouped (for instance in a row of a table).

2) in part 6, the effect of ethylene on auxin transport(ers) should be described! the latter is an essential process in organ formation. 

Some minor things:

Figure legends could do with some more explanation on colors, symbols and abbreviations

line 184: ETR1 = ETHYLENE RESISTANT 1

line 212-214: what does exogenous mean?  if a neighbouring plant within the same cuvette/vessel produces ethylene, that is also exogenous. Also the atmosphere in a vessel cannot be seen as completely controlled.

line 21 additive instead of addictive

line 50: regenerating

line 51: the initial explant

Line 123: according

Line 146: assessed

line 164-165: restructure sentence

line 188: copper

line 252: seem

line 276: a myriad is too widely used. One can say "several" or very maybe "many" The number of studies is quite limited

Line 283: generating

Line 314: mainly

Line 319-321: please restructure the sentence

line 324: auxin

line 328: concerning

throughout the manuscript: use concentrations in µmol/L (µM if wanted) or mg/L. Now things are mixed. 

line 407: decreased ; did not affect

Solanum penelli or pennelli?

line 431: each what?

line 438: sensitivities

line 450: did not affect

line 471: reported for

line 501: remove "on the"

line 511: an increase in 

line 545: of 

Author Response

Answers to the questions of the reviewer 1

Dear Reviewer,

We would like to acknowledge the valuable comments, corrections and suggestions raised by you that helped us to improve the manuscript.

All the comments were taken into account and several sections of the document were rewritten and improved. The manuscript was refocused to better justify the significance of the work and several additional data were included.

Abstract and conclusion were improved and a summary of ethylene effect on Somatic Embryogenesis process was included. Additional studies were also included to reinforce the possible role of ethylene in epigenetic processes.

Our responses to your comments are in bold.

Due to some mistakes in the numbering of references, as well as the replacement of information to other sections and the introduction of new references, the list of references was renumbered. A new version of the manuscript marked up using “Track Changes” was uploaded. 

We hope this new version is now ready to be accepted for publication in plants.

Sincerely,

Jorge Canhoto

Reviewer: 1

The manuscript by Neves et al comprehensively reviews the literature on regeneration and the role of ethylene in this process in plants. They focus on this process and refer to other recent reviews on biosynthesis and signaling of ethylene.

I have the following suggestions:

1) On the SE process, apparently differences in the role of ethylene occur depending on the phase (Embryogenic callus induction, SE growth and further). Could the authors foresee a table or figure in which per phase the role of ethylene is indicated (stimulatory, inhibitory). This sequence of the ethylene effect per phase will depend on the plant species, but it seems some plant tissues have the same mode of ethylene effect sequence, and can thus be grouped (for instance in a row of a table).

Table 3 was included in the manuscript according to the suggestion.

2) in part 6, the effect of ethylene on auxin transport(ers) should be described! the latter is an essential process in organ formation.

The effect of ethylene in auxin transport/transporters was described.

Some minor things:

Figure legends could do with some more explanation on colors, symbols and abbreviations

This suggestion was considered and some improvements on figure captions were made.

line 184: ETR1 = ETHYLENE RESISTANT 1

This correction was made.

line 212-214: what does exogenous mean?  if a neighbouring plant within the same cuvette/vessel produces ethylene, that is also exogenous. Also the atmosphere in a vessel cannot be seen as completely controlled.

The sentence was clarified, and the following sentence was deleted to avoid misunderstandings. However, when we refer to “controlled atmosphere” we wanted to say that in vitro vessels have a more controlled atmosphere compared with plants growing in vivo.

line 21 additive instead of addictive

This correction was made.

line 50: regenerating

This sentence was deleted to make a clear differentiation between SE process and Organogenesis.

line 51: the initial explant

This correction was made.

Line 123: according

This correction was made.

Line 146: assessed

This correction was made.

line 164-165: restructure sentence

The sentence was reformulated.

line 188: copper

This correction was made.

line 252: seem

This correction was made.

line 276: a myriad is too widely used. One can say "several" or very maybe "many" The number of studies is quite limited

This correction was made.

Line 283: generating

This correction was made.

Line 314: mainly

This correction was made.

Line 319-321: please restructure the sentence

The sentence was restructured.

line 324: auxin

This correction was made.

line 328: concerning

This correction was made.

throughout the manuscript: use concentrations in µmol/L (µM if wanted) or mg/L. Now things are mixed.

This correction was made.

line 407: decreased ; did not affect

This correction was made.

Solanum penelli or pennelli?

It was a writing error. Solanum pennelli is correct and the corrections were made.

line 431: each what?

Each was referred to “each treatment”. The sentence was restructured.

line 438: sensitivities

This correction was made.

line 450: did not affect

This correction was made.

line 471: reported for

This correction was made.

line 501: remove "on the"

This correction was made.

line 511: an increase in

This correction was made.

line 545: of

This correction was made.

Reviewer 2 Report

Comments and Suggestions for Authors

The authors describe an important updated review about the molecular pathways involved in the modulation of organogenesis and somatic embryogenesis by ethylene, which can serve not only as valuable basic information to better understand the influence of ethylene modulation in regeneration processes but also to improve micropropagation protocols in plant species for practical uses. Therefore, the article may interest a large audience. The overall presentation is well structured, clear generally, bibliographic reference is robust, and several representative examples are shown on the effect of the ethylene modulator in different regeneration systems in different plant species. However, while it is true that this review focuses on a basic analysis to understand how ethylene affects regeneration processes, this article would have a greater impact if the descriptive analysis of the influence of ethylene modulation on regeneration processes can lead to a joint analysis to suggest future strategies that emphasize essential steps that need to be clarified to resolve the factors that limit the success of micropropagation for practical uses.

Please consider correcting the following points.

Abstract

Should be improved. In addition to the descriptive background of the review, a short proposal of the directions for the future studies should be given.

Conclusion and Future Prospects

Should be improved, including a brief comment about the future strategies that emphasize essential steps that need to be clarified to resolve the factors that limit the success of micropropagation for practical uses.

References

Please go through the whole reference list and check the editorial mistakes in some Journal title abbreviations (e.g. Reference 14, 66, 105, 114 should be abbreviated as Plant Cell Tiss. Organ Cult.).

Minor comments

-L4: correct as Mariana Neves 1, Sandra Correia 1, Carlos Cavaleiro 2 and Jorge Canhoto 1,*

-L5-L13: correct as

 1 Center for Functional Ecology, Department of Life Sciences, University of Coimbra, 5 3000-456 Coimbra, Portugal; mariananevespt@gmail.com (M.N.); sandraimc@ci.uc.pt (S.C.)

 2 University of Coimbra, CIEPQPF, Faculty of Pharmacy, Pólo das Ciências da Saúde, 9 Azinhaga de Santa Comba, 3000-548 Coimbra, Portugal; cavaleir@ff.uc.pt

 * Correspondence: jorgecan@uc.pt; Tel: +00-00-000-0000

-L344: delete space between “[86,87]” and “stresses”

-L506: delete space between “ethylene” and “depends”

-L520: correct “Takn” to “Taken”

-L545: correct “ao” to “of”

-I recommend the use of abbreviations throughout the manuscript, after the explain in the first mention (e.g. somatic embryogenesis in Line 339, L460, L545, L567, L571, should be written as SE).

Author Response

Answers to the questions of the reviewer 2

Dear Reviewer,

We would like to acknowledge the valuable comments, corrections and suggestions raised by you that helped us to improve the manuscript.

All the comments were taken into account and several sections of the document were rewritten and improved. The manuscript was refocused to better justify the significance of the work and several additional data were included.

Abstract and conclusion were improved and a summary of ethylene effect on Somatic Embryogenesis process was included. Additional studies were also included to reinforce the possible role of ethylene in epigenetic processes.

Due to some mistakes in the numbering of references, as well as the replacement of information to other sections and the introduction of new references, the list of references was renumbered.

Our responses are in bold.

A new version of the manuscript marked up using “Track Changes” was uploaded. 

We hope this new version is now ready to be accepted for publication in plants.

Sincerely,

Jorge Canhoto

Reviewer: 2

The authors describe an important updated review about the molecular pathways involved in the modulation of organogenesis and somatic embryogenesis by ethylene, which can serve not only as valuable basic information to better understand the influence of ethylene modulation in regeneration processes but also to improve micropropagation protocols in plant species for practical uses. Therefore, the article may interest a large audience. The overall presentation is well structured, clear generally, bibliographic reference is robust, and several representative examples are shown on the effect of the ethylene modulator in different regeneration systems in different plant species. However, while it is true that this review focuses on a basic analysis to understand how ethylene affects regeneration processes, this article would have a greater impact if the descriptive analysis of the influence of ethylene modulation on regeneration processes can lead to a joint analysis to suggest future strategies that emphasize essential steps that need to be clarified to resolve the factors that limit the success of micropropagation for practical uses.

Please consider correcting the following points.

Abstract

Should be improved. In addition to the descriptive background of the review, a short proposal of the directions for the future studies should be given.

The abstract was improved according to the suggestions.

Conclusion and Future Prospects

Should be improved, including a brief comment about the future strategies that emphasize essential steps that need to be clarified to resolve the factors that limit the success of micropropagation for practical uses.

The conclusion was improved taking in consideration the suggestions.

References

Please go through the whole reference list and check the editorial mistakes in some Journal title abbreviations (e.g. Reference 14, 66, 105, 114 should be abbreviated as Plant Cell Tiss. Organ Cult.).

This correction was made.

Minor comments

-L4: correct as Mariana Neves 1, Sandra Correia 1, Carlos Cavaleiro 2 and Jorge Canhoto 1,*

-L5-L13: correct as

 1 Center for Functional Ecology, Department of Life Sciences, University of Coimbra, 5 3000-456 Coimbra, Portugal; mariananevespt@gmail.com (M.N.); sandraimc@ci.uc.pt (S.C.)

 2 University of Coimbra, CIEPQPF, Faculty of Pharmacy, Pólo das Ciências da Saúde, 9 Azinhaga de Santa Comba, 3000-548 Coimbra, Portugal; cavaleir@ff.uc.pt

 * Correspondence: jorgecan@uc.pt; Tel: +00-00-000-0000

-L344: delete space between “[86,87]” and “stresses”

-L506: delete space between “ethylene” and “depends”

-L520: correct “Takn” to “Taken”

-L545: correct “ao” to “of”

These corrections were made.

-I recommend the use of abbreviations throughout the manuscript, after the explain in the first mention (e.g. somatic embryogenesis in Line 339, L460, L545, L567, L571, should be written as SE).

These corrections were made.

Reviewer 3 Report

In the current review “Modulation of organogenesis and somatic embryogenesis by ethylene: an overview”, Neves et al. provide an updated summary on the effects of ethylene on organogenesis and somatic embryogenesis. Biotic and abiotic stresses affect the role of ethylene in regeneration. Hormonal crosstalk with auxin and cytokinin also modulates the activity of ethylene. Transcription factors, members of the ethylene response factor family, are regulated by ethylene and take a major part in the induction of somatic embryogenesis.   

The following are some comments, issues, and typos found in the manuscript that hopefully, the authors find helpful.

Line 21. Additive

Line 101. The statement is a bit misleading. The absolute levels of EIN2 don't change, only EIN2 localization is affected. Instead, try to say something similar to this:

... (EIN2). The lack of phosphorylation promotes the cleavage of the EIN2 C-end that moves from the ER to the nucleus, promoting the stabilization of the EIN3/EIL transcription factors, which leads to...

Line 188. Copper

Line 179 to 195. Do you think it is okay to move this entire paragraph to section 2 where the signaling pathway is described?
As a reader, it feels a bit forced to have it here. The authors may think the opposite. This paragraph provides more depth into the receptors than the previous one into the ACS/ACO enzymes. They seem to be a bit unbalanced.

Line 197. Does the application of Ag ions also cause some off-target effects? Toxicity? Introduce here this because it will come later in the manuscript. How does compare 1-MCP to Ag? Which one is better? Cleaner? Triggers fewer side effects?

Line 230. Figure 2, the ethylene absorbents compounds are not present in the figure.

Line 296 to 304. Can you elaborate more on this statement? It is the same molecule, but are both precesses (ethylene biosynthesis and DNA methylation) somehow connected? Is there a direct link between ethylene and DNA methylation? If not, amend the statement.

Line 302. Again, elaborate more, you need to further explain your statements. Is there a direct effect of ethylene on the activity of specific methyltransferases that target DNA/RNA? If not, modify your statement, it can be misleading.

Line 354. Please, review your entire references list.
Here, reference 78 is misplaced. Reference number 78 is Martins, J.F.; Correia, S.I.; Canhoto, J.M. Somatic embryogenesis induction and plant regeneration in strawberry tree (Arbutus 774 unedo L.). In Methods in Molecular Biology; Germana, M., Lambardi, M., Eds.; Humana Press: New York, NY, USA, 2016; Volume 775 1359, pp. 329–339. 10.1007/978-1-4939-3061-6_14 

The only reference in your list with Biddington as a first author is number 6.

Line 354. Usually, the numbering of the references matches their appearance in the text. It is not the case in this manuscript.

Line 371. Here, does ETH mean ethylene or ethephon? Not clear. I think it is ethephon, right?

Line 378. When all these concentrations and treatments are provided, it is a bit confusing when different units are used. It would be better to choose one, either μM or mg.L-1, and always use the same for consistency. 

Line 461. A period is missing after development.

Line 520. Taken

Line 522. Is it possible to overcome the altered ethylene signal transduction pathway in the mentioned mutants by just adding more auxin to the media? Would that induce SE in these mutants? Elaborate a bit more on this conclusion.

Line 545. case of alfalfa

Line 555. Methylglyoxal bis(guanylhydrazone is only mentioned in the tables. Could you please, introduce this compound with the rest of the inhibitory compounds in the corresponding section? 

Line 590. Conclusion section.

In this section, it would be great to summarize the review in a figure with the effects of ethylene on the regeneration processes. Authors could focus on some of the examples provided in the text and include the effects of other hormones and transcription factors involved.

...

Author Response

Answers to the questions of the reviewer 3

Dear Reviewer,

We would like to acknowledge the valuable comments, corrections and suggestions raised by you that helped us to improve the manuscript.

All the comments were taken into account and several sections of the document were rewritten and improved. The manuscript was refocused to better justify the significance of the work and several additional data were included.

Abstract and conclusion were improved and a summary of ethylene effect on Somatic Embryogenesis process was included. Additional studies were also included to reinforce the possible role of ethylene in epigenetic processes.

Due to some mistakes in the numbering of references, as well as the replacement of information to other sections and the introduction of new references, the list of references was renumbered. A new version of the manuscript marked up using “Track Changes” was uploaded. 

Our responses are in bold.

We hope this new version is now ready to be accepted for publication in plants.

Sincerely,

Jorge Canhoto

Reviewer: 3

In the current review “Modulation of organogenesis and somatic embryogenesis by ethylene: an overview”, Neves et al. provide an updated summary on the effects of ethylene on organogenesis and somatic embryogenesis. Biotic and abiotic stresses affect the role of ethylene in regeneration. Hormonal crosstalk with auxin and cytokinin also modulates the activity of ethylene. Transcription factors, members of the ethylene response factor family, are regulated by ethylene and take a major part in the induction of somatic embryogenesis.  

The following are some comments, issues, and typos found in the manuscript that hopefully, the authors find helpful.

Line 21. Additive

This correction was made.

Line 101. The statement is a bit misleading. The absolute levels of EIN2 don't change, only EIN2 localization is affected. Instead, try to say something similar to this:

... (EIN2). The lack of phosphorylation promotes the cleavage of the EIN2 C-end that moves from the ER to the nucleus, promoting the stabilization of the EIN3/EIL transcription factors, which leads to...

The sentence was rewritten. Concerning the EIN2 levels, phosphorylation of EIN2 leads to its ubiquitination, inhibition of CTR1 (when ethylene is present) leads to a consequent increase in EIN2 levels (since it is no longer ubiquitinated). Thus, we reformulated the sentence and included the remaining suggestions.

Line 188. Copper

This correction was made.

Line 179 to 195. Do you think it is okay to move this entire paragraph to section 2 where the signalling pathway is described?

As a reader, it feels a bit forced to have it here. The authors may think the opposite. This paragraph provides more depth into the receptors than the previous one into the ACS/ACO enzymes. They seem to be a bit unbalanced.

We agree. This correction was made, and the paragraph was moved to section 2.

Line 197. Does the application of Ag ions also cause some off-target effects? Toxicity? Introduce here this because it will come later in the manuscript. How does compare 1-MCP to Ag? Which one is better? Cleaner? Triggers fewer side effects?

Some references were added to state some phytotoxic effects of silver. The non-toxicity of 1-MCP was confirmed and added to the text.

Line 230. Figure 2, the ethylene absorbents compounds are not present in the figure.

This correction was made, and the original Figure 2 was changed.

Line 296 to 304. Can you elaborate more on this statement? It is the same molecule, but are both precesses (ethylene biosynthesis and DNA methylation) somehow connected? Is there a direct link between ethylene and DNA methylation? If not, amend the statement.

Line 302. Again, elaborate more, you need to further explain your statements. Is there a direct effect of ethylene on the activity of specific methyltransferases that target DNA/RNA? If not, modify your statement, it can be misleading.

The sentence was restructured, and some important references were added, such as direct involvement of EIN2-C in histone acetylation.

Line 354. Please, review your entire references list.

Here, reference 78 is misplaced. Reference number 78 is Martins, J.F.; Correia, S.I.; Canhoto, J.M. Somatic embryogenesis induction and plant regeneration in strawberry tree (Arbutus 774 unedo L.). In Methods in Molecular Biology; Germana, M., Lambardi, M., Eds.; Humana Press: New York, NY, USA, 2016; Volume 775 1359, pp. 329–339. 10.1007/978-1-4939-3061-6_14

The only reference in your list with Biddington as a first author is number 6.

Line 354. Usually, the numbering of the references matches their appearance in the text. It is not the case in this manuscript.

We had in fact some mistakes in reference numbering. This correction was made and now the references match their appearance in the text.

Line 371. Here, does ETH mean ethylene or ethephon? Not clear. I think it is ethephon, right?

ETH means ethephon. The first time ethephon was referred (Section 4.) the respective abbreviation (ETH) was indicated and we kept using it throughout the manuscript.

Line 378. When all these concentrations and treatments are provided, it is a bit confusing when different units are used. It would be better to choose one, either μM or mg.L-1, and always use the same for consistency.

This correction was made.

Line 461. A period is missing after development.

This correction was made.

Line 520. Taken

This correction was made.

Line 522. Is it possible to overcome the altered ethylene signal transduction pathway in the mentioned mutants by just adding more auxin to the media? Would that induce SE in these mutants? Elaborate a bit more on this conclusion.

Extra information was added, and the statement was clarified.

Line 545. case of alfalfa

This correction was made.

Line 555. Methylglyoxal bis(guanylhydrazone is only mentioned in the tables. Could you please, introduce this compound with the rest of the inhibitory compounds in the corresponding section?

This information was placed in Section 4.

Line 590. Conclusion section.

In this section, it would be great to summarize the review in a figure with the effects of ethylene on the regeneration processes. Authors could focus on some of the examples provided in the text and include the effects of other hormones and transcription factors involved.

We take this suggestion into consideration and a summary, and a possible model for ethylene effect is now illustrated in Figure 3